# Investigations on Melt Flow Rate and Tensile Behaviour of Single, Double and Triple-Sized Copper Reinforced Thermoplastic Composites

**DOI:** 10.3390/ma14133504

**Published:** 2021-06-23

**Authors:** Balwant Singh, Raman Kumar, Jasgurpreet Singh Chohan, Sunpreet Singh, Catalin Iulian Pruncu, Maria Luminita Scutaru, Radu Muntean

**Affiliations:** 1Department of Mechanical Engineering, Chandigarh University, Mohali 140413, India; balwant.e1941@cumail.in (B.S.); raman.me@cumail.in (R.K.); jasgurpreet.me@cumail.in (J.S.C.); 2Department of Mechanical Engineering, National University of Singapore, Singapore 117575, Singapore; snprt.singh@gmail.com; 3Design, Manufacturing and Engineering Management, University of Strathclyde, Glasgow G4 0LN, UK; 4Department of Mechanical Engineering, Transilvania University of Brasov, 500036 Brasov, Romania; 5Department of Civil Engineering, Transilvania University of Brasov, 500036 Brasov, Romania; radu.m@unitbv.ro

**Keywords:** polymer matrix composites, Melt Flow Index, Nylon 6, ABS, copper particles

## Abstract

Thermoplastic composite materials are emerging rapidly due to the flexibility of attaining customized mechanical and melt flow properties. Due to high ductility, toughness, recyclability, and thermal and electrical conductivity, there is ample scope of using copper particles in thermoplastics for 3d printing applications. In the present study, an attempt was made to investigate the Melt Flow Index (MFI), tensile strength, and electrical and thermal conductivity of nylon 6 and ABS (acrylonitrile butadiene styrene) thermoplastics reinforced with copper particles. Thus, the experiments were conducted by adding different-sized copper particles (100 mesh, 200 mesh, and 400 mesh) in variable compositions (0% to 10%) to ABS and nylon 6 matrix. The impact of single, double, and triple particle-sized copper particles on MFI was experimentally investigated followed by FTIR and SEM analysis. Also, the tensile, electrical, and thermal conductivity testing were done on filament made by different compositions. In general, higher fluidity and mechanical strength were obtained while using smaller particles even with higher concentrations (up to 8%) due to improved bonding strength and adhesion between the molecular chains. Moreover, thermal and electrical conductivity was improved with an increase in concentration of copper particles.

## 1. Introduction

The use of polymer matrix composites has been increasing day-by-day due to vast engineering applications of these materials [1,2]. Polymers are preferred for fabrication of light-weight components involving lower manufacturing cost and lead time. Due to unique characteristics of polymers such as higher toughness, resilience, corrosion resistance, flexibility, and strength to weight ratio, they have completely replaced metals for certain application areas [3,4]. Polymer matrix composites have been developed to induce customized characteristics and further enhance the scope of applications. Metal particles are added to polymer matrices to improve stiffness, fatigue resistance, wear resistance, specific strength, and conductivity of products. Metal-based polymer matrix composites are in high demand in a variety of fields, including automotive, aerospace, civil, electronics, communications, sports, marine, military, building, and various household items [5,6]. Recent investigations have demonstrated that, in general, the physical, mechanical, electrical, and thermal properties of polymers are improved with the addition of reinforcements [7]. Tekce et al. [8] reported an increase in thermal conductivity of polymers with the addition of copper particles and found a higher role of particle size, shape, and concentration. Copper-based polymer composites are essentially used by electronic industries as higher electrical and thermal conductivity is achieved [9]. Although numerous technologies are available for production of polymer composites, Fused Deposition Modeling (FDM) has evolved as a promising technique, as products with customized designs can be prepared. Furthermore, desired properties can be induced through FDM by varying the manufacturing conditions and concentration of reinforcements while maintaining minimum production cost and time [10].

The raw material for FDM is a thermoplastic filament that can be prepared through extrusion process. Boparai et al. [11] explored the possibilities of using polymer composite filament through extrusion for rapid prototyping and tooling. It was emphasized that the melt flow rate of composite filament must match with virgin material to ensure uniform extrusion during the FDM process. The significant variation in melt flow rate results in practical problems such as nozzle clogging, filament breakage, porous structure, non-uniform heating etc. Thus, the study of melt flow rate and thermal and mechanical strength of filaments is desirable during the development of polymer composites using FDM [12].

Numerous studies on strength and fluidity of composites prepared through screw extrusion and FDM process have been reported by researchers. Hamzah et al. [13] investigated the mechanical, electrical, and thermal characteristics of Acrylonitrile Butadiene Styrene (ABS) reinforced with Copper Ferrite (CuFe_2_O_4_) with three different weight proportions (8%, 11% and 14%). So, it was concluded that the specimens that were reinforced with 14% CuFe_2_O_4_ particles exhibited 135% improvement in tensile strength. The study suggests considerable interlocking of CuFe_2_O_4_ filler with ABS matrix, which enhanced its bonding strength and hence provided resistance to tensile loading. Due to greater resistance to plastic deformation, a 14% rise in ABS hardness was measureded with the addition of 14% CuFe_2_O_4_. The thermal conductivity was improved by 93% and also electrical conductivity was significantly improved by adding CuFe_2_O_4_ in ABS. Shahmirzadi et al. [14] studied the rheological and morphological properties of composites prepared by addding copper particles to Polypropylene matrix. It was reported that the melt flow rate of feedstock filament decreased to one third as the concentration of copper particles increased from 50% to 60%. In the case of feedstock, this fluidity reduction occurred due to improper mixing of copper powder and binder. It was concluded that for better mixing of powder and binder, time and temperature of mixing are two critical parameters. Moreover, the particle size of the copper powder played an important role during rheological testing. Better electrical and thermal conductivity was achieved with lower particle size due to uniform distribution of copper even at higher concentrations. Isa et al. [15] reported an increase in melt flow rate of ABS composites with the addition of 50 µm copper reinforcements. It was found that melting temperature and mixing conditions significantly affect the fluidity of composites. Nikzad et al. [16] studied the mechanical and thermal properties of specimens manufactured by 40% addition of iron particles in ABS matrix. The thermal and mechanical properties of composites were enhanced as compared to virgin materials, which highlighted the importance of metallic fillers. Singh et al. [17] investigated the torsion and tensile behavior of ABS-clay nanocomposites with morphological and rheological characterizations. ABS nano-composites were manufactured in a twin-screw extruder machine by mixing them with the various levels of nanoclay (Cloisite 30B) followed by characterization. It was concluded that the addition of nanoclay to ABS improved the modulus of elasticity and modulus of rigidity under torsional load.

Despite extensive literature being available on composites prepared through extrusion, there is a need to investigate the impact of different copper particle sizes and concentrations on melt flow rate, and tensile, thermal, and electrical properties of composite filament. The present study was conducted to explore the possibility of using copper-reinforced 3d-printed thermoplastic polymer for electronic components. Recent studies have reported an increase in electrical and thermal conductivity of polymers with the addition of copper particles [18,19]. Hence, the customized electronic components can be prepared from copper-based polymer composites using FDM process. Also, the impact of combinations of different particles sizes on melt flow, mechanical, and morphological properties of composite filament have not yet been explored. ABS and nylon 6 are selected as matrix material, as these are extensively used by electronics industry for making casings, enclosures, cooling fan blades, wearable devices, and cabinets. Higher thermal conductivity of composite material ensures faster dissipation, which further prevents overheating of electronic parts. Moreover, the electrical conductivity induced by copper doping finds vast applications in sensors and metamaterials. The findings of this study would help to select suitable composition and combination of copper particles for preparing composite filament with desired mechanical, thermal, and electrical properties, and flow rate. The composite filament can be further used for fabrication of customized electronic products.

## 2. Materials and Methods

Nylon 6 and ABS are used as base materials to prepare composite filaments during present experimentation. Nylon 6 is a thermoplastic material that is chemically resistive and abrasion resistive. Its glass transition temperature is 47 °C, melting temperature is 220 °C, and density is 1.31 g/cm^3^. Because of these properties, the nylon 6 is generally used in the production of gears and bearings used in the automotive industry. ABS is a low-cost engineering thermoplastic material that has properties such as impact resistance and rigidity, but it can be easily machined, processed, and thermoformed. ABS provides a good balance of resistance to heat, dimensional stability, high tensile strength, high stiffness, and insulating properties [20]. The majority of natural ABS resins are translucent to opaque, but they can be manufactured in transparent grades and almost any color can be pigmented [21]. For certain outdoor applications, general-purpose grades can work, but constant sun exposure causes the color change. The constant exposure also decreases material gloss, toughness, fracture toughness, and modulus of elasticity. The ABS melting temperature ranges from 221–240 °C, and the density is 1.07 g/cm^3^ [22]. Copper, having a melting point of 1083 °C, has many advantageous properties such as corrosion resistivity and strong electrical and thermal properties [23]. For the present research, the nylon 6 and ABS material was supplied by Batra Polymers Pvt. Ltd., Ludhiana, India. On the other hand, copper particles of different sizes were purchased from Shri Manak Copper Pvt. Ltd., Jaipur, India.

Nylon 6 and ABS (acrylonitrile butadiene styrene) were used as a matrix material while copper particles of three different sizes (100 mesh, 200 mesh, 400 mesh) were used as a reinforcement material (see Figure 1). Some of the properties of copper particles have been displayed in Table 1.

The density of 100 mesh, 200 mesh, and 400 mesh copper particles is 1.32 g/cc, 1.22 g/cc, and 1.29 g/cc respectively. The selection of different particle sizes was done based on previous literature as particle shape and size have a significant effect on product characteristics [8,24]. Using different combinations, the composite material with customized strength and flow rate can be prepared as per requirement. This finds practical applications where weight and cost must be optimized as per strength requirement to save product cost. Since copper is costly as compared to polymers, the product cost can be reduced by controlling particle size and combination.

Figure 2 describes the detailed methodology adopted to complete the present experimental work. Initially, the nylon 6 material and ABS (acrylonitrile butadiene styrene) thermoplastics were selected for the research work with the different compositions of copper particles of variable (100 mesh, 200 mesh, and 400 mesh) sizes. After the selection of materials, composite mixtures were prepared using different sizes and different proportions. In the present work, first of all, single-particle size compositions were prepared using the weighing machine in which three composite materials were prepared each with ABS and nylon 6 as the matrix. Secondly, double particle size compositions were prepared by mixing each thermoplastic with two different sizes of copper particles. Thirdly, triple particle size compositions were prepared by mixing equal proportions of three sizes of copper particles. In this research work, the notations are given to each type of composition as shown in Table 2. After the preparation of all compositions, the Melt Flow Rate was measured with the help of Melt Flow Indexer through which the fluidity of the material could be ascertained. The tensile testing of different compositions was performed with the help of the Universal Testing Machine. At last, the characterization was done with the help of the Scanning Electron Microscope (SEM) and Fourier Transform Infrared Spectroscopy (FTIR).

### 2.1. Melt Flow Index Testing

Melt Flow Index (MFI) of polymer composites was measured to ascertain their flow rate, which is necessary for its usability in FDM. This test simply measures the weight ratio of the flow rate of polymers or polymer composites over a given time and temperature range. ASTM D1238 is just one particular standard that has been used for Melt Flow Index (MFI) testing [29]. This method is one of the most common methods used by different industries when describing the exact fluidity of the material. The experiments were performed in this study using the melt indexer machine (MIM) [30]. Initially, the MIM was preheated at 220 °C for around 30 min and afterward, 5–8 g of the mixture was loaded in the MIM. A piston load of 2.180 kg was mounted on the piston rod, and a stopwatch was used to read 25.4 mm in length. A filament was extruded in the vertical direction according to gravity flow. During the present experimentation work, the MIM (see Figure 3) used to test the melt flow rate of polymer composite was Model: TC-SW-20-E supplied by Shanta Engineering Pvt. Ltd., Pune, India. With a temperature range of ambient to 400 °C, the MIM was equipped with a Microprocessor-based PID Temperature Controller having 0.1 °C resolution.

### 2.2. Universal Testing Machine

To acquire the tensile properties of polymer composites such as peak load, break load, peak elongation, and breaking elongation, the universal testing machine was used [31]. In the present experiment, we used universal testing machine manufactured by Shanta Engineering Pvt. Ltd., Pune, India (Model: Standard Twin Column) as shown in Figure 4. This machine has a pulley belt system that allows for the selection of different speeds. It also has the ability to mount numerous load cells as well as grips and fixtures for various tests and materials such as rubber, plastics, cables, leather, paper, plywood, and polymer composites. A motorized test frame is driven by twin ball screws featuring dual columns and designed to apply tension ranging from 2 N to 50 kN. The tensile testing was performed following ASTM D4018-17 standards using composite filament of diameter 1.75 mm and length 90 mm.

### 2.3. FTIR and SEM Testing

In the present investigation, Fourier transform infrared spectroscopy (FTIR) was used to characterize the composites based upon their bonding behaviour. This can be confirmed by the amount of infrared rays absorbed by the solid, liquid, or gas when placed in the spectrometer. The Equations (1)–(5) are used for finding the frequencies *ν_1_* and *ν_2_*.
*d* = *n*λ_1_*and d* = (*n* + *1*)*λ_2_*(1)
*λ_1_* = *d/n and λ_2_* = *d*/(*n* + *1*)(2)
*ν_1_* = *1*/*λ_1_ and ν_2_* = *1*/*λ_2_*(3)
*ν_1_* = *n/d and ν_2_* = (*n* + *1*)/*d*(4)
*ν_2_* − *ν_1_* = *1/d*(5)
where: *d* represents the path differences,

*λ_1_* and *λ_2_* represents the wavelengths,

*n and (n + 1)* shows the cycles,

And *ν_1_* and *ν_2_* represent the frequencies.

The frequency of the spectrum ranges from 550 cm^−1^ to 4000 cm^−1^. These frequency spectrums are divided into five different stages, which are classified as: (i) 2800–4000 cm^−1^ (ii) 2000–2500 cm^−1^, (iii) 1500–2000 cm^−1^, (iv) 1000–1500 cm^−1^, and (v) 550–1000 cm^−1^. Fundamentally, the FTIR shows the functional group in the form of peaks produced on the graphs. The frequency ranges from 2800–4000 cm^−1^, forming the bonds like C–H, N-H, and O-H having the spectrums sp^3^, sp^2^, and sp respectively. The sp^3^ spectrum has the frequency of 2850 cm^−1^, sp^2^ having the frequency of 3000 cm^−1^, whereas the frequency of sp is 3100 cm^−1^. So, it means the value increases while moving from sp^3^ towards sp. Now, in the second stage, the frequency varies from 2000–2500 cm^−1^. In this range, C≡N bond forms at the frequency of 2250 cm^−1^, C≡C bond forms at the frequency of 2100 cm^−1^, and C=C=C bond forms at the frequency of 2150 cm^−1^. The frequency in-between the 1500–2000 cm^−1^ in which C=C varies from 1400–1500 cm^−1^, C=N varies from 1500–1600 cm^−1^, and C=O has a frequency in the range of 1680–1810 cm^−1^. In between the frequencies from 1000–1500 cm^−1^ C-C, C-N, C-O exists, whereas from 550–1000 cm^−1^ C-I, C-Br and C-Cl exist [32]. In the present experiment, the infrared spectrum absorption or emission on different compositions of filaments was evaluated on Fourier-transform infrared spectroscopy (FTIR) machine supplied by PerkinElmer Inc., Waltham, MA, USA (Model: PerkinElmer spectrum 2).

Scanning Electron Microscope (SEM) was utilized to visualize the surface and fracture behavior of composites. SEM is a medium for visualizing micro and nano-sized details by impinging a high-energy electron beam on the material surface [33]. The SEM apparatus was supplied by Jeol Ltd., Tokyo, Japan (Model: IT500) having high vacuum and low vacuum resolutions, with Filament Electron gun, and 10–650 Pa Pressure adjustment.

### 2.4. Electrical and Thermal Conductivity Testing

The electrical conductivity testing was performed using insulation resistance tester (Model: S1-522/2) supplied by Megger Ltd., Dover, Kent, United Kingdom having a range of 50 V to 5000 V (see Figure 5). This instrument has provision to alter voltage and current loops, allowing accurate measurement of resistance regardless of the voltage applied. Since this method is relatively simple, it does not destruct the samples. In the present research work, the resistivity for different compositions was tested with Megger tester from which the conductivity was measured by using the relationship between resistivity and conductivity. The relations between resistance (R), electrical resistivity (ρ), and conductivity (σ) are shown by Equations (6) and (7).
ρ = R × Area/Length(6)
σ = 1/ρ(7)

Thermal conductivity was calculated by the Lee’s Disc method [34]. In Lee’s Disc Method, the sample is placed in-between two brass discs and maintained at temperature T1 and T2 as shown in Figure 6. A steam chamber with intake and exit ports is placed above the discs. It also has circular holes for thermocouple insertion, which helps to describe a material’s ability to conduct the heat. A steady state is quickly obtained while steam is passing through vessel, which is in cylindrical shape. Heat carried through the testing sample is equivalent to heat emitted from the Lee’s disc at steady state. Finally, thermal conductivity is measures using Equation (8).
(8)$$K=\frac{\mathrm{mC}\;\left(\frac{\mathrm{dT}}{\mathrm{dt}}\right)X}{A\;\left(T_{2}-T_{1}\right)}$$

K = Thermal conductivity of the specimen

A = Cross sectional area,

T_2_ − T_1_ = Temperature difference across the specimen.

X = Thickness of the specimen

m = mass of disc

C = Specific heat of disc (for brass-376 J/kg °C)

dT/dt = difference in temperature w.r.t time

## 3. Results and Discussion

### 3.1. Melt Flow Properties

During melt flow testing, significant variation in fluidity was experienced due to particle size and concentration. Initially, nylon 6 material was mixed separately with *A, B, C* sized copper particles at different proportions to check the melt flow rate of particular compositions. Firstly, *A* was mixed with the nylon 6 material in three different proportions i.e., 1%, 2.5% and 5%, which gave composition *A1*, *A2* and *A3*. By using the weighing machine, particles were mixed and then three readings of melt flow rate were taken using MIM by putting 2.180 kg weight at 220 °C, and the mean value of melt flow rate was finally considered. The results show that with an increase in the quantity 100 mesh copper particles in the nylon 6 materials, the melt flow rate was decreasing continuously. Similarly, the composition with *B* and *C* particles with nylon 6 material at different ratios was prepared. The data collected during this experiment are shown in Table 3 in which the mean and standard deviation values were retrieved for each combination. It was observed that the standard deviation of all the composites was different, which occurs due to difference in mean values for each experiment.

It can be noted that an increase in the concentration of reinforcements in Nylon6 decreased the MFI of polymer composites irrespective of copper particle size. This is attributed to the reduced fluidity of the polymer matrix due to the addition of metal particles. The polymer granules and metal particles were not properly attached, and there was a lack of proper bonding between polymer and metal.

#### 3.1.1. Composition of ABS with Single Copper Particles

ABS material was mixed with copper particles at different proportions to check the flow rate at every stage by using the 2.180 kg ram weight and 220 °C temperature of MIM. In the first place, 1% *A* (Copper) and 99% (ABS) named as *A4* were mixed together to investigate the mean flow rate, which was calculated as 2.243 gm/10 min. Similarly, the mixtures *A5* and *A6* were prepared. While increasing the copper particles in ABS, the melt flow rate was decreasing as experienced in the case of Nylon6.

In the second place, *B4* was prepared using the weighing machine after which the melt flow rate was calculated as 1.881 gm/10 min. This was followed by the preparation and testing of *B4, B5, B6, B7*, and *B8* compositions. It was noticed that the melt flow rate was increasing from 1% to 5%, but afterwards, with further addition of 200 mesh copper particles, the melt flow rate was decreasing.

Thirdly, when *C4* compositions were prepared and tested, it was observed that as the percentage of copper particles was increased, the melt flow rate also increased, which was not experienced in previous compositions of both ABS and Nylon6. The ABS mixed with 400 mesh size copper was smoothly extruded out from the nozzle due to higher fluidity rate. It was examined that the fluidity increases while increasing the copper percentage, which means the intermixing capacity is better for 400 mesh Copper particles with ABS. Thus, while increasing the 400 mesh copper particles in ABS, the melt flow rate correspondingly increases. The size of these particles is appropriate for ABS molecules to attach and form a uniform mixture. All the detailed readings are displayed in Table 4 in which mean and standard deviation value are discussed, and where it is shown that the most deviated value was of *B5* composition, whereas the least deviated value was of *B7* composition. This also occurs due to difference in mean values of all the compositions.

#### 3.1.2. Composition of ABS with Double Particle Size

The double copper particle means equal mixing of two different-sized copper particles in the polymer material. Furthermore, Nylon6 was not used to prepare double copper size compositions as MFI was severely reduced in previous studies. The reduced MFI of filaments would not be beneficial for 3D printing applications. Firstly, in this experimental work, mixing of 1% (*A* and *B*) of equal proportion with 99% ABS was done. The mean value of melt flow rate of *(A + B)4* was obtained as 2.42 gm/10 min. After that, the mean value of melt flow rate of *(A + B)9* and *(A + B)10* was 2.34 gm/10 min and 2.27 gm/10 min respectively. So, it was observed that the melt flow rate was decreasing while increasing the mixture of *100* and *200* mesh copper particles with ABS.

The mean values of MFI of *(B + C)4, (B + C)9*, and *(B + C)10* were 1.99 gm/10 min, 1.7 gm/10 min, and 2.25 gm/10 min respectively. From these results, it was deduced that the melt flow rate was decreasing up to *3%* loading of 200 and 400 mesh copper particles. However, with the further addition of double-sized particles, there was an abrupt increase in MFI, which indicates that *400* mesh copper particles help to increase the fluidity.

After testing of *(A + C)4, (A + C)9*, and *(A + C)10* mixtures, it was found that while increasing the (*A* and *C*) particles concentration ratios in ABS, the melt flow rate also increased. The combination of 100 mesh with 400 mesh copper particles improved the fluidity of the mixture. The detailed measurements of MFI of double particle size-reinforced ABS are shown in Table 5.

#### 3.1.3. Composition of ABS and Nylon 6 with Triple Particle Size

In this experimental study, three different sizes of copper particles were mixed in equal proportions with ABS and nylon 6 to check the melt flow rate. The mean value of melt flow rate of *(A + B + C)11* was 18.25 gm/10 min. Furthermore, the mean values of MFI for *(A + B + C)12* and *(A + B + C)13* were 28.27 gm/10 min and 30.73 gm/10 min respectively. It can be noticed from the above mean values that melt flow rate was increasing while increasing the mixture of (*A, B* and *C*) copper particles with nylon 6.

Furthermore, the testing of triple-sized copper particles mixed with ABS revealed a similar scenario. The mean values of MFI of compositions (*A + B + C)9*, *(A + B + C)10*, and (*A + B + C*) were measured as 1.885 gm/10 min, 2.28 gm/10 min, 2.45 gm/10 min respectively. So, from the above discussion, it was observed that while increasing the mixture of (*A, B*, and *C*) copper particles with ABS, the melt flow rate was also increased. Table 6 shows the full description of values obtained during experimentation in which the mean value and standard deviation values are displayed. From the standard deviation value, the most dispersed value was obtained in *(A + B + C)11* material composition while the least dispersed value was obtained at *(A + B + C)10* composite material. The efficacy of 400 mesh-sized copper particles is highlighted by these investigations. The smaller size of copper reinforcements showed better adherence and mixing with thermoplastic materials [35]. This resulted in ease of flow during the molten state, which further improved the MFI.

The Scanning Electron Microscope (SEM) images of composites having different ratios of copper particles with ABS are shown in Figure 7.

The SEM images in Figure 7a represents large voids in the case of composites with 2.5% copper particles with 97.5% ABS, while Figure 7b represents the lesser voids as copper concentration was increased to 5%. Furthermore, in Figure 7c, the packed structure having 8% copper particles is visualized. This indicates that with an increase in the quantity of copper particles in ABS polymers, there is continuous reduction of voids and porosity in the structure. This reduction in voids improved the flow rate of composite material, and hence it can be deduced that a higher concentration of copper particles with smaller particle size is suitable for 3d printer.

### 3.2. Tensile Properties

After the melt flow rate testing of different composite materials, the tensile testing of extruded filaments with different compositions was performed for comparison. The length and diameter of the filaments were fixed as 90 mm and 1.75 mm respectively for the tensile testing for uniformity. In this tensile testing process, the peak load (N), peak elongation (mm), break load (N), break elongation (mm), strength at peak (MPa), strength at break (MPa), % elongation at peak, % elongation at break, Young’s modulus, and Poisson’s ratio were calculated [36]. The tensile testing was done only for ABS as filaments prepared with Nylon6 were not found suitable for 3D printing applications. ASTM D4018-17 is a standard testing technique used for testing tensile properties of composite filaments with diameter 1.75 mm and length 90 mm.

#### 3.2.1. Tensile Testing of Single Copper Particle with ABS

First of all, the Tensile Testing was done for the filaments that have the composition of *A4, A9, and A10*. It was concluded that tensile properties were maximum for the composition with 3% copper particles, whereas the strength at peak was maximum for the composition having 6% reinforcements. On the other hand, the maximum strength at break was achieved in case of composition *A4*. It was concluded that the ultimate tensile strength was increasing while the percentage of *A* Copper particles was added up to 3%. After that, there was a sudden drop in ultimate tensile strength with further addition of 100 mesh copper particles in ABS. The whole data is shown in tabular form in Table 7. Also, the stress–strain graphs are displayed in Figure 8a–c. Tensile testing was done on multiple specimens for a particular composition, which is represented in each figure.

Secondly, the tensile testing was done for the compositions *B4, B9, B10, B7*, and *B8*, which were prepared using 200 mesh copper particles. It was found that the maximum values of tensile properties were obtained for the composition of 8% copper and 92% ABS. The maximum value of percentage elongation at break was reached during the composition of 10% copper, while the break elongation was maximum for 1% reinforcement. It was concluded that the ultimate tensile strength was decreasing continuously while increasing the percentage of copper particles with ABS, but it was maximum at 8% concentration. Table 8 shows the overall readings and stress–strain diagram for overall compositions in Figure 9a–e.

Thirdly, the tensile testing was done for the compositions *C4, C9*, and *C10*. It was observed that break elongation was maximum for 1% copper particles, whereas the peak load, break load, strength at peak, and strength at break were highest for 3% copper particles. However, the peak elongation and elongation at peak were maximum for the composition of 6% copper particles. It was observed that the ultimate tensile strength was maximum for composition 3% reinforcement while ultimate tensile strength was nearly the same for 1% and 6% compositions. All of the information is tabulated in Table 9 and the stress–strain curves shown in Figure 10a–c.

It was noticed that the maximum ultimate tensile strength was obtained in the composition of 8% copper particles (200 mesh) with 92% ABS, whereas the minimum ultimate tensile strength was achieved with a composition of 10% copper.

#### 3.2.2. Tensile Testing of Double Copper Particle with ABS

In this case, the tensile testing was done for the filaments that were made with mixing of two different sizes of copper particles with ABS at various ratios. Initially, the compositions *(A + B)4, (A + B)9, (A + B)10* were prepared using 100 and 200 mesh copper particles. It was observed that peak load, break load, strength at peak, and % elongation at peak were maximum at the composition with 3% mixture of copper particles. Whereas the peak elongation, break elongation, and % elongation at break were maximum in 1% reinforcement. It must be further noted that the ultimate tensile strength was the same for composites prepared using 1% and 3% copper particles. A sudden decrease in ultimate tensile strength was noted while adding more double-sized copper particles (*A* and *B*) in ABS. Table 10 shows the tensile properties and Figure 11a–c represents the stress–strain curves.

Now, the filaments prepared from *(B + C)4, (B + C)9, (B + C)10* were tested for comparison. It was experienced that peak load, peak elongation, break load, strength at peak, strength at break, and % elongation at peak were highest for 3% addition. Conversely, for 6% addition, the break elongation and % elongation at break were maximum. It was inferred that the ultimate tensile strength was lowest for 6% and maximum for 3% copper reinforcement. Table 11 shows the exact readings for all compositions and Figure 12a–c shows the stress–strain curves for particular compositions.

Thirdly, the double-sized compositions of (*A + C)4, (A + C)9*, and (*B + C*) were prepared and filaments were used for testing. It was observed that peak load, peak elongation, break load, break elongation, strength at peak, strength at break, % elongation at peak, and % elongation at break were maximum for the composition of the 1% copper. It can be further noted that the ultimate tensile strength was decreasing continuously with the addition of double-size copper particles (100 and 400 mesh) in ABS. All of the data in tabulated form is shown in Table 12 with stress–strain curves in Figure 13a–c.

Overall, it was confirmed that the highest ultimate tensile strength was achieved in double-sized copper particles with a composition of *3%* (200 and 400 mesh) copper particles and *97%* ABS, whereas the lowest ultimate tensile strength was achieved using the composition of *6%* (100 and 400 mesh) copper particles and 94% ABS.

#### 3.2.3. Tensile Testing of Triple Copper Particle with ABS

The mixture of all three sizes of copper particles at different proportions was prepared with ABS as the matrix. It was concluded that peak properties were minimum during 1% addition whereas maximum for 3% addition of copper particles. The data are shown in Table 13 with stress–strain curves in Figure 14a–c.

The above analysis indicates that there was a decrease in ultimate tensile strength while adding triple–sized copper particles from 1% to 3%. Further addition of copper particles (upto 6%) increased the ultimate tensile strength.

A variation in stress–strain curves of different replications with same composition was noticed. This can be attributed to a variation in diameter of filament during extrusion. The filament with 1.75 mm diameter was extruded with the tolerance of ±0.5 mm. Hence, the overall filament diameter of filament was achieved within the range 1.70–1.80 mm. Moreover, this diameter was not uniform along the length, which resulted in variation in mass and volume. Islam et al. [37] experienced similar dimensional variation in filaments, which resulted in significant variation in tensile behavior of T700 carbon fiber. The variation in stress–strain curves during the same experiments with different replications was demonstrated by Rane et al. [38]. During the tensile testing, crazing phenomenon occurred at multiple regions, which resulted in significant disagreement between stress–strain curves. The defects in 3d-printed tensile test specimens and changes in manufacturing conditions tend to exhibit significant differences in mechanical strength [39,40]. In the present experiment, the impact of different copper particle sizes and their combinations were tested to determine tensile strength. At the initial stages of new material development, such problems of dimensional variations were evident and also reported by previous studies during the extrusion of polymers and polymer composites [41,42]. The variation in diameter of filament can be reduced by studying and controlling the critical parameters during extrusion. During commercialization of new material, the noise factors can be removed, which would improve the consistency in tensile test measurements.

### 3.3. Material Characterization

#### 3.3.1. FTIR Analysis

FTIR method was used to characterize the functional groups obtained at different frequencies while adding different-sized copper particles in ABS and nylon 6. The IR spectra of base material depends upon the shape and size of the reinforced particles [43]. FTIR of 200 mesh-sized copper particles with 2.5% addition in nylon 6 is shown in Figure 15a. With an addition of 2.5% copper in nylon 6 material, we get an initial sharp peak at a frequency of 2920.99 cm^−1^, which indicates CH_2_ asymmetric stretching. Another absorption was noted at 1724.87 cm^−1^, which represents the stretching vibration of C=O bond. Similar absorption was experienced by Charles et al. [44] after the addition of glass fiber in nylon 6. The smaller peaks at frequencies of 1633.13 cm^−1^ and 1536.40 cm^−1^ highlight the presence of Amide I and Amide II. However, the lower reflectance was noticed as compared to pure Nylon 6, which occurred due to a larger size of copper particles. The aggregations caused by large-sized particles resulted in higher absorbance. Also, the peak observed at 696.84 cm^−1^ indicates the higher contribution of copper particles during shift in wavelength and absorption rate. This peak in spectrum was also reported by Betancourt-Galindo et al. [45] during the synthesis of copper nanoparticles. This aggregation of large-sized particles also reduced the MFI of composite prepared with 200 mesh copper particles (*B2*) as compared to 100 mesh *(A2)* and 200 mesh *(C2)* (see Table 3). The FTIR of samples with *5%* 400 mesh copper particles in Nylon6 is shown in Figure 15b. The frequencies obtained are 2927.03 cm^−1^ and 3292.37 cm^−1^, which indicates the sp and sp^2^ have the chemical bond C-H. Because most organic compounds feature C-H bonds, a relevant guideline is that absorption between 2850 and 3000 cm^−1^ is caused by sp3 C-H stretching, whereas absorption beyond 3000 cm^−1^ is caused by sp2 C-H stretching or sp C-H stretching if the wavelength is near 3300 cm^−1^. As the s nature of the C-H bond increased, so did its strength. It also has two peaks with sharp ends achieved at frequencies of 1636.27 cm^−1^ and 1535.05 cm^−1^ making a C=N. The C=N double bond is around twice as strong as a C-N single bond, while a C-N triple bond has roughly the same strength as a double bond. The 400 mesh-sized copper particles show comparatively greater adhesion and bonding with polymer chains. This phenomenon validates the higher MFI and tensile strength of composites with 400 mesh-sized copper particles as compared to 100 and 200 mesh sizes.

FTIR for ABS with different ratios of copper particles are shown in Figure 16 and Figure 17. In the case of *C5* and C6 composition, the sharp peak at 2921.27 cm^−1^ represents the sp^2^ spectrum having a chemical bond of C-H. C-H stretches only below 3000 cm^−1^ in compounds that do not have a C=C bond. As the s nature of the C-H bond increased, so did its strength. The depth of the potential energy well associated with a C-H is measured by bond strength. The stiffness of a bond is a measurement of the amount of energy required to compress or stretch it. The firmer bond is frequently associated with a deeper potential energy surface, despite the fact that these are separate qualities. The peak at a frequency near 1500 cm^−1^ represents C=N chemical structure.

For C mixture, three peaks were examined from which one peak was 2920.04 cm^−1^, which means sp^2^ spectrum making C-H chemical bonding. The other two peaks were nearer to 700 cm^−1^, making C-Cl bonding.

Lastly, when *C7* and *C8* were tested, then the highest peak was only obtained at the frequency 698.15 cm^−1^, from which it was concluded that while increasing the frequency, the value of ν (nu) was decreasing. FTIR images for these mixtures are shown in Figure 15.

The FTIR analysis of composite samples reveals the presence of C-H chemical bonding, which is also supported by the higher values of tensile strength and MFI as compared to pure materials. This phenomenon also supports the significant increase in MFI and tensile strength of composites prepared with 400 mesh copper particles. The smaller size of particles ensures uniform dispersion and chemical bonding between the amorphous molecular chains as compared to larger particles [46,47]. The optimum composition and size of copper particles improved the rheology and mechanical stability of samples, and the phenomenon was supported by FTIR results.

#### 3.3.2. SEM Analysis

SEM analysis was done to study the fracture behavior and failure of composite filament during tensile loading. Figure 18a shows the SEM image of 2.5% C copper with 97.5% nylon 6, which reflects that the material has a higher elasticity with no voids. These images support the higher fluidity and mechanical strength of composites prepared with 400 mesh copper particles. Sierra-Avila et al. [48] also performed similar experimentation on composites of nylon 6 and copper nanoparticles. It was investigated through SEM analysis that the composition having 0.10% copper particles in nylon 6 had a high rate of dispersion rate whilst showing spherical and long-shape nanowires in it. Also, the composition of nylon 6 and copper nanoparticles resulted in the highest tensile strength and lowest elongation at break. Unal et al. [49] also experienced a similar phenomenon during SEM analysis, in which the addition of fillers such as wollastonite, kaolin, powder, and glass particles reduced the voids and increased the flexibility of polymer composite. In Figure 18b, the SEM image was done for the composition of 1% *B* copper and 99% ABS material, which represents a lot of voids and stiffer material. The lower values of strength and MFI of ABS samples can be explained by these images. Figure 18c represents the mixture with 6% *B* copper and 94% ABS. The voids were reduced with an increase in copper percentage. As a result, the material became comparatively more elastic as compared to 1% loading. Figure 18d shows the mixture of 10% copper and 90% ABS where the voids are absent, leading to high elasticity and tensile strength. So, it was evident from the SEM images that while increasing the 400 mesh copper percentage in ABS material, the elasticity increased with a reduction in voids.

### 3.4. Electrical and Thermal Conductivity

After melt flow and mechanical strength analysis, the electrical and thermal conductivity of copper-reinforced polymer composites was investigated to highlight their functionality for vast engineering applications. The electrical conductivity testing was done for the compositions of *B5, B6, B7*, and *B8* having 2.5%, 5%, 8%, and 10% copper particle reinforcement respectively. The filaments samples with diameter 0.203 m and length 0.05 mm were prepared for electrical conductivity testing. The readings obtained from the testing confirm that the resistivity is decreasing as the percentage of copper is increased in ABS polymer. Conversely, the electrical conductivity of polymer composite is increased with copper concentration. The measurements were obtained after the testing, and hence the graphical representation of electrical conductivity is shown Figure 19. The increase in copper particle concentration assists the flow of electricity. This property of composite polymer can be used for the development of functional prototypes, 4d materials, and smart materials.

During the thermal conductivity measurement, disc-shaped samples with a mass of 0.25 kg and area 0.007853 m^2^ were prepared through FDM. A variation in thermal conductivity of composites was noticed for every composition. Figure 20 plots a relationship between copper particle concentration and thermal conductivity of samples. An increase in particle concentration in ABS improves the ability to conduct heat as more particles are dispersed, which act as heat carriers.

## 4. Conclusions

The melt flow rate, mechanical, morphological, electrical, and thermal properties of copper-reinforced ABS and nylon 6 filaments were investigated to highlight their applicability for 3d printing applications. An attempt was made to investigate the scope of using 3D-printed copper-reinforced polymers for fabrication of different industrial application parts from which the higher mechanical, thermal, and electrical conductivity can be obtained. The copper particles with three different sizes were used as reinforcement and the impact of different combinations and compositions was investigated. It was noticed that MFI firstly increased with the addition of copper particles up to a particular point, after which constant fluidity could be obtained, which could be used to prepare feedstock filament. The melt flow rate and tensile strength improved by using small-sized copper particles. Overall, it was noticed that the fluidity and tensile strength obtained within the range of 5–8% addition of copper particles (200 and 400 mesh) is suitable for making the filament for FDM using a screw extruder. It was evident from FTIR images that this composition made the interfacial chemical bond, which improves bonding strength. Also, the samples prepared with 10% copper reinforcements exhibited the highest thermal and electrical conductivity. The findings of this can help to prepare feedstock filament for FDM process to manufacture customized electronic components with desired mechanical strength. An efficient heat transfer can be ensured by using copper particles in polymers, which would also enhance the mechanical stability of components.

## Figures and Tables

**Figure 1 materials-14-03504-f001:**
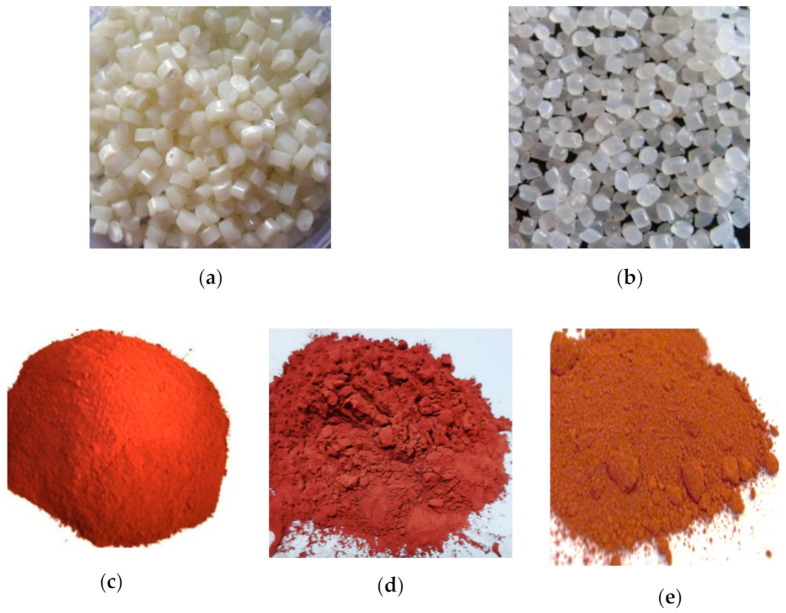
Images of: (**a**) ABS, (**b**) nylon 6, (**c**) 100 mesh copper powder, (**d**) 200 mesh copper powder, and (**e**) 400 mesh copper powder.

**Figure 2 materials-14-03504-f002:**
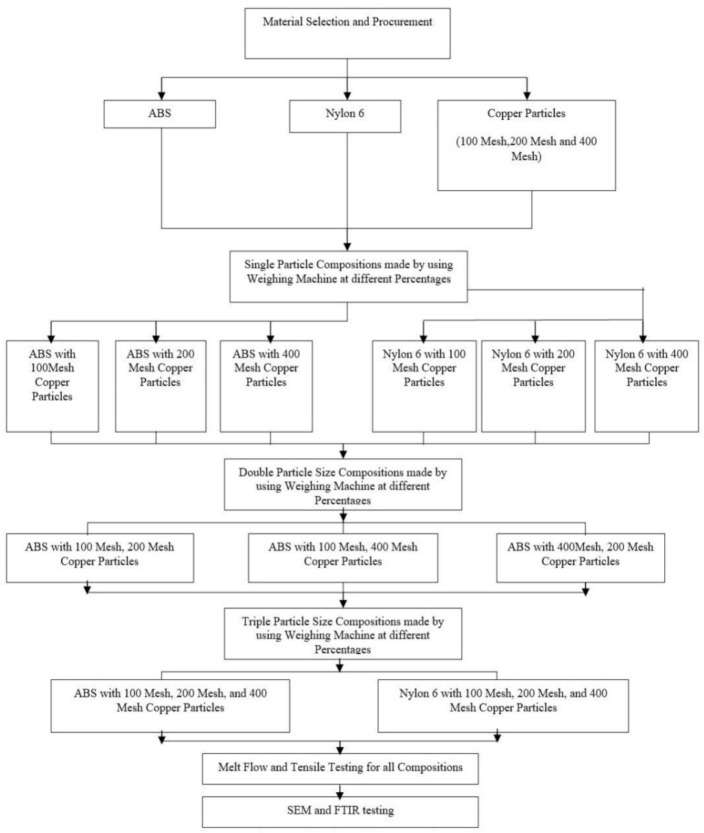
Flow chart of present study.

**Figure 3 materials-14-03504-f003:**
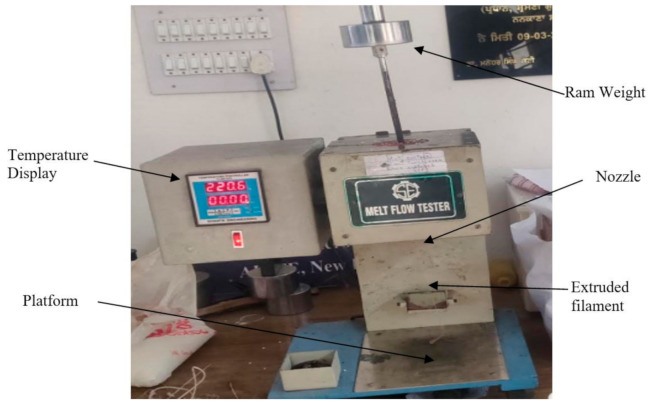
Melt indexer machine.

**Figure 4 materials-14-03504-f004:**
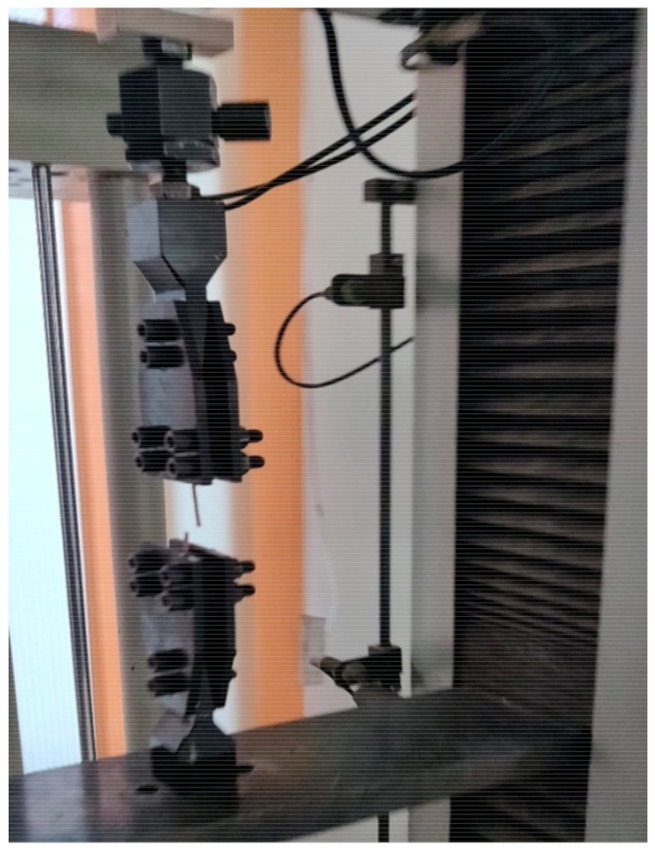
Universal testing machine.

**Figure 5 materials-14-03504-f005:**
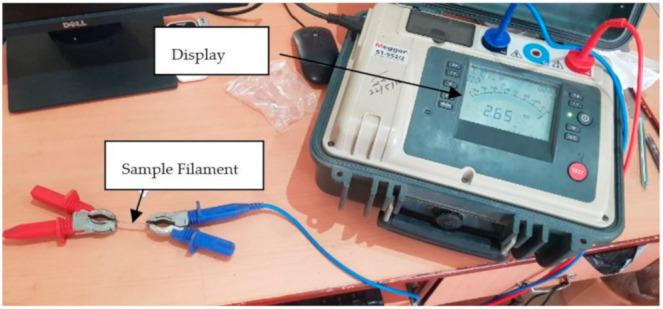
Electrical resistivity testing apparatus.

**Figure 6 materials-14-03504-f006:**
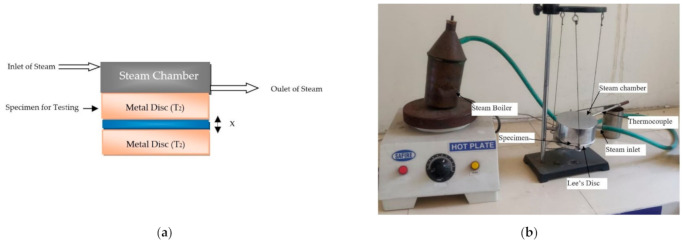
(**a**) Thermal conductivity measurement principle; (**b**) Lee’s Disc Experimental Setup.

**Figure 7 materials-14-03504-f007:**
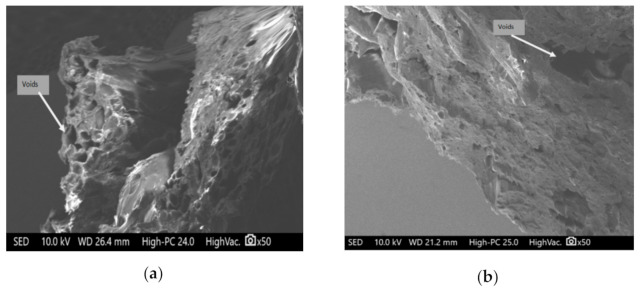
SEM images of (**a**) *B5*, (**b**) *B6*, and (**c**) *B8*.

**Figure 8 materials-14-03504-f008:**
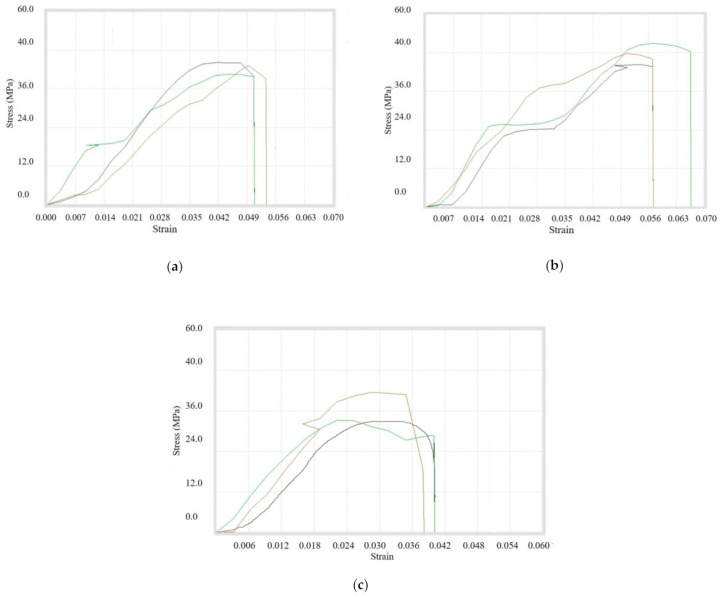
Stress–strain curves for (**a**) *A4*, (**b**) *A9*, and (**c**) *A10* filament samples.

**Figure 9 materials-14-03504-f009:**
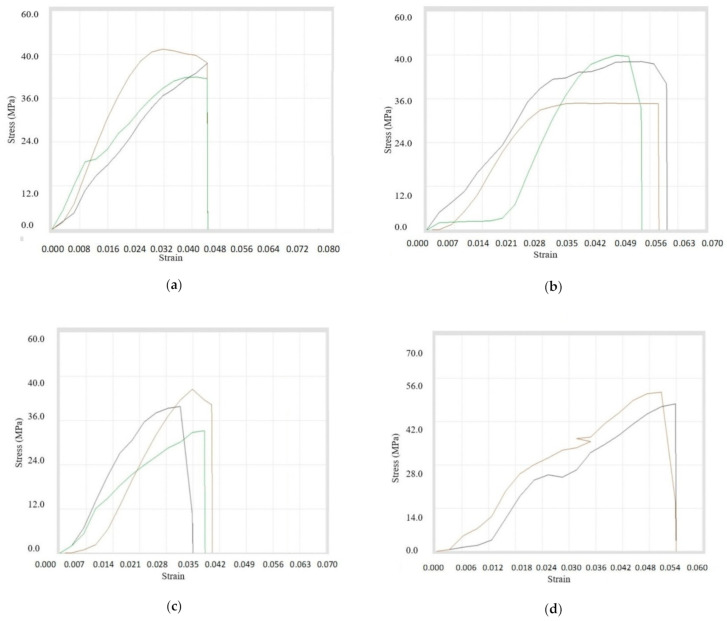
Stress–strain curves for (**a**) *B4*, (**b**) *B9*, (**c**) *B10*, (**d**) *B7*, and (**e**) *B8* composite samples.

**Figure 10 materials-14-03504-f010:**
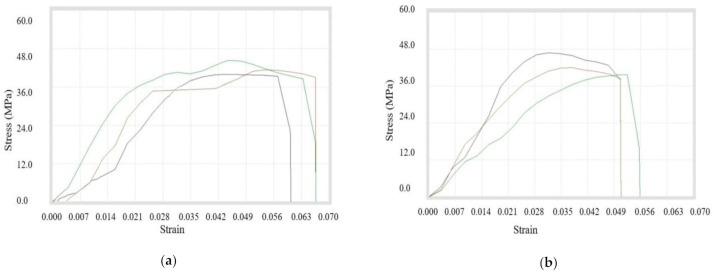
Stress–strain Curves for (**a**) *C4*, (**b**) *C9*, and (**c**) *C10* filament samples.

**Figure 11 materials-14-03504-f011:**
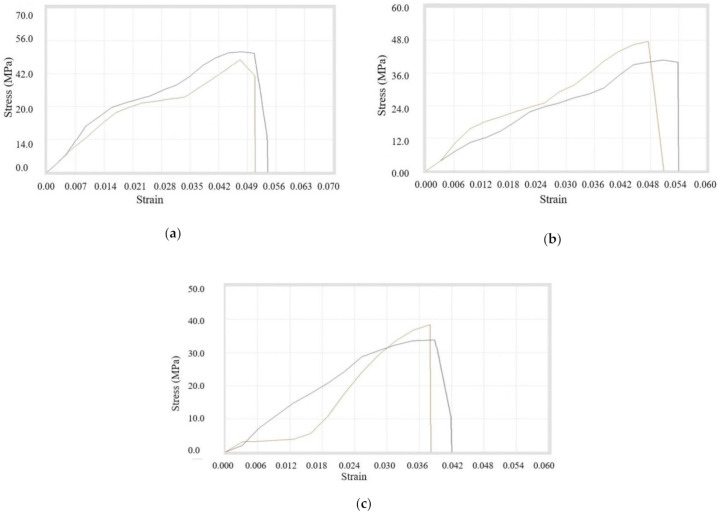
Stress–strain curves for (**a**) *(A + B)4*, (**b**) *(A + B)9*, (**c**) *(A + B)10* composite samples.

**Figure 12 materials-14-03504-f012:**
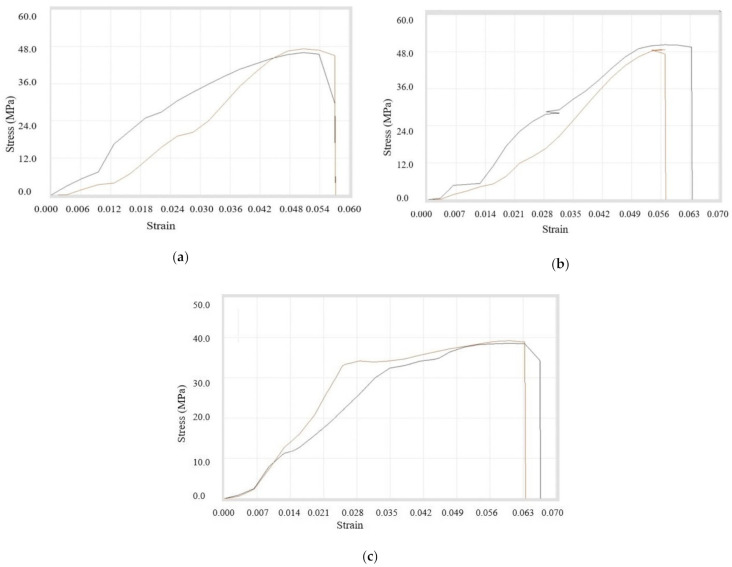
Stress–strain curves for (**a**) *(B + C)4*, (**b**) *(B + C)9*, (**c**) *(B + C)10* composite samples.

**Figure 13 materials-14-03504-f013:**
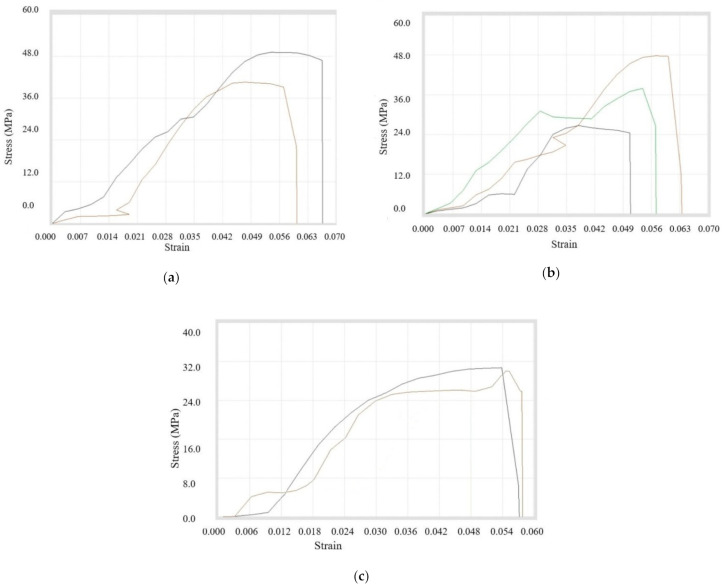
Stress–strain Curves for (**a**) *(A + C)4*, (**b**) *(A + C)9, and* (**c**) *(B + C)10* filament samples.

**Figure 14 materials-14-03504-f014:**
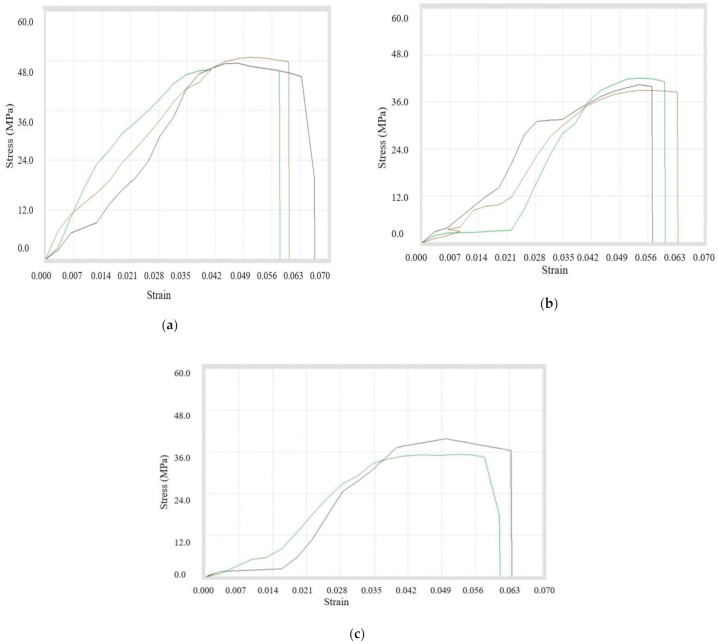
Stress–strain curves for (**a**) *(A + B + C)4*, (**b**) *(A + B + C)9*, and (**c**) *(A + B + C)10* filament samples.

**Figure 15 materials-14-03504-f015:**
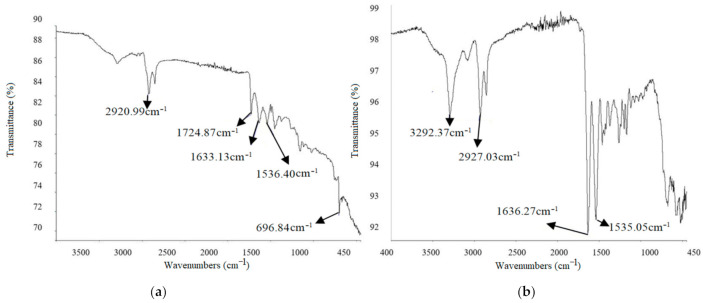
FTIR graphs of Nylon composites: (**a**) *B2* and (**b**) *C3*.

**Figure 16 materials-14-03504-f016:**
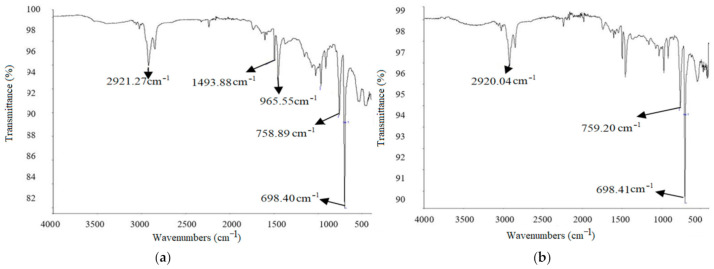
FTIR graphs of ABS composites: (**a**) *C5* and (**b**) *C6*.

**Figure 17 materials-14-03504-f017:**
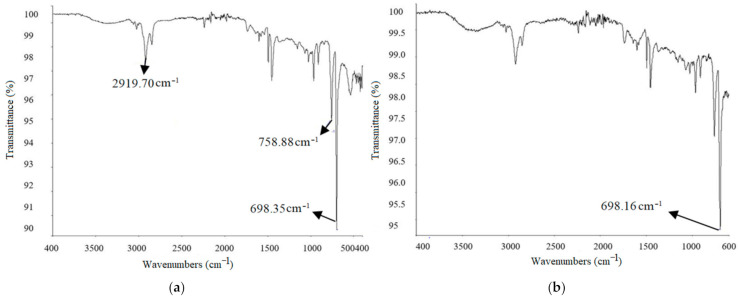
FTIR graphs of ABS composites (**a**) *C7* and (**b**) *C8*.

**Figure 18 materials-14-03504-f018:**
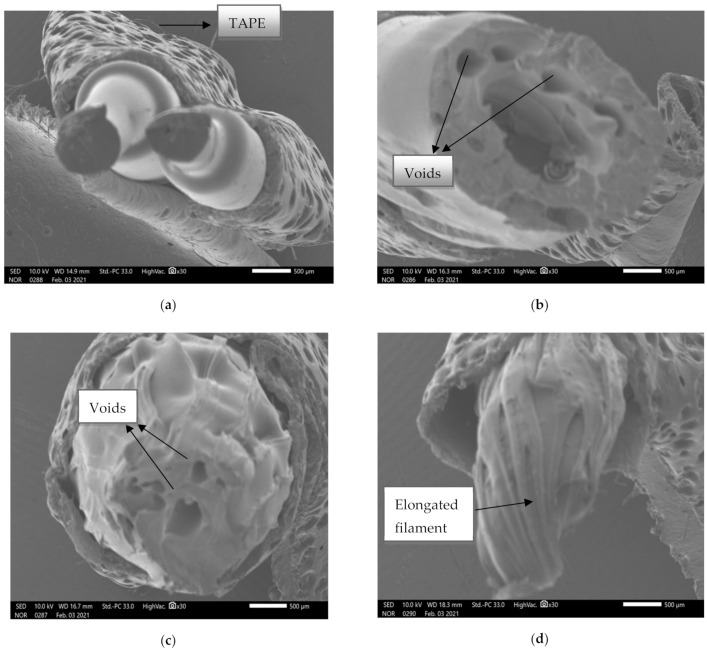
SEM Images of (**a**) *C2*, (**b**) *B4*, (**c**) *B10*, and (**d**) *C8*.

**Figure 19 materials-14-03504-f019:**
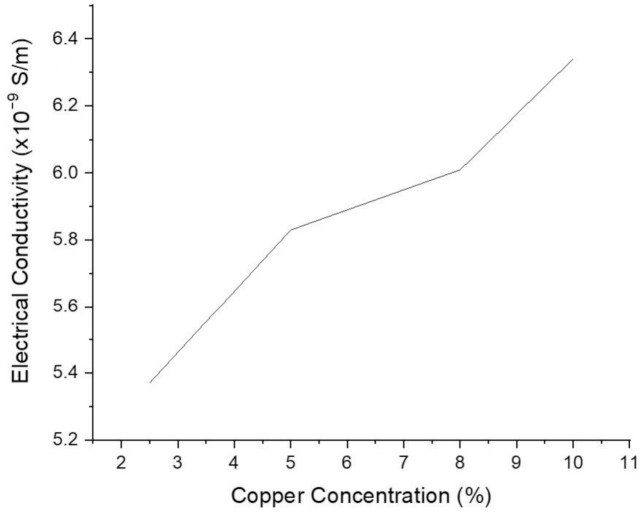
Electrical conductivity of copper-reinforced polymer composites.

**Figure 20 materials-14-03504-f020:**
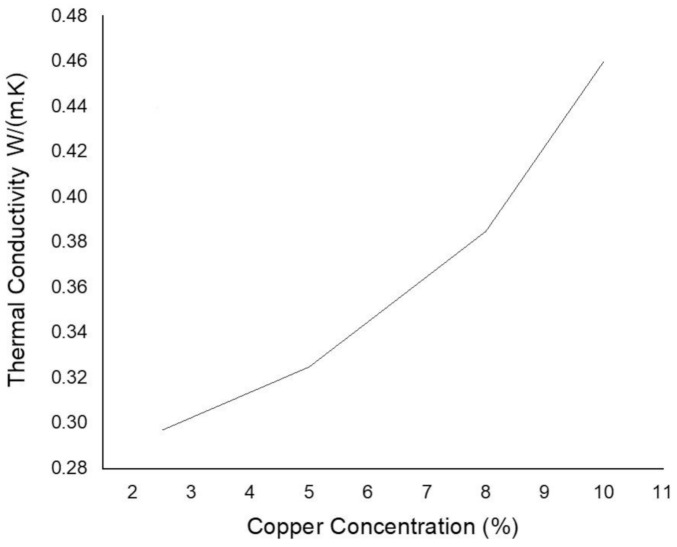
Thermal conductivity of copper-reinforced polymer composites.

**Table 1 materials-14-03504-t001:** Properties of copper particles.

Characteristics	Test Method	Specification	Test Results
Apparent Density (g/cc)	IS4848 [25]	1.0–1.4	1.32
Sieve Analysis (%)	IS460 [26]	100%	100%
Particle Shape	Light Optical Microscopy	Dendritic	Dendritic
Color	Visual Inspection	Reddish Brown	Reddish Brown
Flow Rate (Sec/50 g)	IS 4840 [27]	50–60	60
Chemical Analysis (%)	IS 440 [28]	99.8 Min	99.9

**Table 2 materials-14-03504-t002:** Nomenclature of different compositions.

Serial Number	Particle Size and Compositions	Notation
1.	100 mesh Copper Particles	*A*
2.	200 mesh Copper Particles	*B*
3.	400 mesh Copper Particles	*C*
4.	1% Copper and 99% Nylon 6	*1*
5.	2.5% Copper and 97.5% Nylon 6	*2*
6.	5% Copper and 95% Nylon 6	*3*
7.	1% Copper and 99% ABS	*4*
8.	2.5% Copper and 97.5% ABS	*5*
9.	5% Copper and 95% ABS	*6*
10.	8% Copper and 92% ABS	*7*
11.	10% Copper and 90% ABS	*8*
12.	3% Copper and 97% ABS	*9*
13.	6% Copper and 94% ABS	*10*
14.	3% Copper and 97% Nylon 6	*11*
15.	6% Copper and 94% Nylon 6	*12*
16.	8% Copper and 92% Nylon 6	*13*
17.	8% Copper and 92% ABS	*14*

**Table 3 materials-14-03504-t003:** MFI of nylon 6 reinforced with copper particles.

Material Composition	1st Reading	2nd Reading	3rd Reading	Mean	Standard Deviation
*A* Copper particles with nylon 6 (gm/10 min)					
*A1*	7.41	6.175	6.536	6.707	0.635
*A2*	5.905	5.445	6.725	6.025	0.648
A3	4.54	4.57	4.055	4.388	0.289
*B* Copper particles with nylon 6 (gm/10 min)					
*B1*	4.11	4.732	4.526	4.456	0.316
*B2*	3.805	4.695	4.126	4.208	0.450
B3	2.515	1.675	1.985	2.058	0.424
*C* Copper particles with nylon 6 (gm/10 min)					
*C1*	9.28	7.315	8.225	8.273	0.983
*C2*	5.795	5.738	5.765	5.766	0.028
*C3*	3.68	3.76	3.65	3.696	0.056

**Table 4 materials-14-03504-t004:** MFI of ABS reinforced with single particle size copper.

Material Composition	1st Reading	2nd Reading	3rd Reading	Mean	Standard Dev.
*A* Copper particles with ABS (gm/10 min)					
*A4*	2.17	2.24	2.32	2.243	0.075
*A5*	1.565	1.785	1.68	1.676	0.110
*A6*	1.47	1.61	1.715	1.598	0.122
*B* Copper particles with ABS (gm/10 min)					
*B4*	1.715	1.79	2.138	1.881	0.225
*B5*	2.48	2.17	2.03	2.226	0.23
*B6*	2.64	2.35	2.38	2.456	0.159
*B7*	2.385	2.46	2.32	2.388	0.070
*B8*	2.29	2.365	2.22	2.291	0.072
*C* Copper particles with ABS (gm/10 min)					
*C4*	1.475	1.635	1.65	1.586	0.096
*C5*	1.575	1.755	1.68	1.67	0.090
*C6*	2.115	1.715	1.905	1.911	0.200

**Table 5 materials-14-03504-t005:** MFI of ABS reinforced with double particle size copper.

Material Composition	1st Reading	2nd Reading	3rd Reading	Mean	Standard Dev.
*A + B* Copper particles with ABS (gm/10 min)					
*(A + B)4*	2.45	2.405	2.41	2.421	0.024
*(A + B)9*	2.32	2.325	2.38	2.341	0.033
*(A + B)10*	2.245	2.325	2.26	2.276	0.042
*B + C* Copper particles with ABS (gm/10 min)					
*(B + C)4*	2.075	1.885	2.03	1.996	0.099
*(B + C)9*	1.68	1.775	1.648	1.701	0.066
*(B + C)10*	2.27	2.245	2.25	2.255	0.013
*A + C* Copper particles with ABS (gm/10 min)					
*(A + C)4*	1.935	1.735	2.305	1.991	0.289
*(A + C)9*	2.09	2.085	1.825	2	0.151
*(A + C)10*	2.265	2.375	2.29	2.31	0.057

**Table 6 materials-14-03504-t006:** MFI of ABS and nylon 6 with triple particle size copper.

Material Composition	1st Reading	2nd Reading	3rd Reading	Mean	Standard Dev.
Mixture of *A, B and C* copper particles with nylon 6 (gm/10 min)					
*(A + B + C)11*	19.975	18.515	16.285	18.258	1.858
*(A + B + C)12*	30.25	26.65	27.91	28.27	1.826
*(A + B + C)13*	31.76	29.90	30.55	30.73	0.943
Mixture of *A, B and C* copper particles with ABS (gm/10 min)					
*(A + B + C)9*	1.785	1.865	2.005	1.885	0.111
*(A + B + C)10*	2.245	2.31	2.285	2.28	0.032
*(A + B + C)14*	2.64	2.35	2.38	2.456	0.159

**Table 7 materials-14-03504-t007:** Tensile testing of *A* copper and ABS at different proportions.

Material Composition	Peak Load (N)	Peak Elongation (mm)	Break Load (N)	Break Elongation (mm)	Strengthat Peak (MPa)	Strengthat Break (MPa)	Elongation at Peak (%)	Elongation at Break (%)	Young’s Modulus (MPa)	Poisson’s Ratio
*A4*	135.93	2.72	122.34	3.2	43.3	51.1	4.3	5.3	795.753	1.6352
*A9*	149.83	3.10	134.85	3.6	47.7	42.9	5.0	6.3	769.354	1.5444
*A10*	112.43	2.53	101.19	2.8	49.7	44.7	4.3	4.7	982.212	1.2566

**Table 8 materials-14-03504-t008:** Tensile testing of *B* copper and ABS at different proportions.

Material Composition	Peak Load (N)	Peak Elongation (mm)	Break Load (N)	Break Elongation (mm)	Strength at Peak (MPa)	Strength at Break (MPa)	Elongation at Peak (%)	Elongation at Break (%)	Young’s Modulus (MPa)	Poisson’s Ratio
*B4*	145.10	2.34	130.59	3.3	46.2	41.6	4.0	5.3	987.179	1.372
*B9*	134.97	2.66	121.47	3.2	43.0	38.7	4.7	5.0	808.270	1.238
*B10*	126.47	2.47	113.82	2.7	40.3	36.2	4.0	4.3	815.789	0.977
*B7*	155.90	2.95	140.31	3.23	49.65	44.685	5	5	841.525	1.443
*B8*	110.30	3.04	99.27	3.135	35.13	31.615	5	5.5	577.796	0.991

**Table 9 materials-14-03504-t009:** Tensile testing of *C* copper and ABS at different proportions.

Material Composition	Peak Load (N)	Peak Elongation (mm)	Break Load (N)	Break Elongation (mm)	Strength at Peak (MPa)	Strengthat Break (MPa)	Elongation at Peak (%)	Elongation at Break (%)	Young’s Modulus (MPa)	Poisson’s Ratio
*C4*	131.70	2.72	118.53	3.9	41.9	37.7	4.3	6.7	770.220	1.4703
*C9*	134.67	2.34	121.20	3.3	42.9	38.6	4.0	5.3	916.666	1.2738
*C10*	131.03	2.91	117.93	3.8	41.7	37.6	4.7	6.7	716.494	1.4288

**Table 10 materials-14-03504-t010:** Tensile testing of *A + B* copper and ABS at different proportions.

Material Composition	Peak Load (N)	Peak Elongation (mm)	Break Load (N)	Break Elongation (mm)	Strength at Peak (9 MPa)	Strengthat Break (MPa)	Elongation at Peak (%)	Elongation at Break (%)	Young’s Modulus (MPa)	Poisson’s Ratio
*(A + B)4*	138.25	3.14	124.43	3.6	44.0	39.6	5.0	6.0	700.636	1.4256
*(A + B)9*	138.95	2.95	125.06	3.1	44.3	39.8	5.1	5.0	750.847	1.2338
*(A + B)10*	118.40	2.57	106.56	2.8	37.7	33.9	4.5	4.5	733.463	0.9492

**Table 11 materials-14-03504-t011:** Tensile testing of *B* and *C* and ABS at different proportions.

Material Composition	Peak Load (N)	Peak Elongation (mm)	Break Load (N)	Break Elongation (mm)	Strength at Peak (9 MPa)	Strength at Break (MPa)	Elongation at Peak (%)	Elongation at Break (%)	Young’s Modulus (MPa)	Poisson’s Ratio
*(B + C)4*	128.68	2.76	115.81	2.9	41.0	36.9	4.8	4.8	742.753	1.0701
*(B + C)9*	155.35	3.42	139.82	3.6	49.5	44.5	6.0	6.0	723.684	1.602
*(B + C)10*	122.05	3.42	109.85	3.9	38.9	35.0	6.0	6.5	568.713	1.365

**Table 12 materials-14-03504-t012:** Tensile testing of *A* and *C* copper and ABS at different proportions.

Material Composition	Peak Load (N)	Peak Elongation (mm)	Break Load (N)	Break Elongation (mm)	Strengthat Peak (MPa)	Strengthat Break (MPa)	Elongation at Peak (%)	Elongation at Break (%)	Young’s Modulus (MPa)	Poisson’s Ratio
*(A + C)4*	141.40	3.04	127.26	3.8	45.0	40.5	5.0	6.5	740.131	1.539
*(A + C)9*	117.63	2.85	105.87	3.4	37.5	33.7	4.3	5.7	657.894	1.145
*(A + C)10*	88.90	2.95	80.01	3.1	28.3	25.5	5.0	5.5	479.661	0.790

**Table 13 materials-14-03504-t013:** Tensile testing of triple-sized particle reinforcements at different proportions.

Material Composition	Peak Load (N)	Peak Elongation (mm)	Break Load (N)	Break Elongation (mm)	Strength at Peak (MPa)	Strength at Break (MPa)	Elongation at Peak (%)	Elongation at Break (%)	Young’s Modulus (MPa)	Poisson’s Ratio
*(A + B + C)4*	148.83	2.66	133.95	3.4	47.4	42.7	4.3	5.7	890.977	1.451
*(A + B + C)9*	127.27	3.10	114.54	3.6	40.5	36.5	5	6	653.225	1.314
*(A + B + C)10*	136.10	2.41	122.49	3.0	43.3	39.0	4	4.7	898.340	1.17

## Data Availability

The data presented in this study are available upon request from the corresponding author.

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
