# Peer review of "Investigations on Melt Flow Rate and Tensile Behaviour of Single, Double and Triple-Sized Copper Reinforced Thermoplastic Composites"

_materials, 2021, doi:10.3390/ma14133504_

Round 1

Reviewer 1 Report

The authors of the submitted manuscript investigated the mechanical behavior of thermoplastics reinforced with copper particles for 3D printing applications in electronics. The Introduction can be shortened especially at the beginning, i.e. the description what are the composites seems to be trivial for the community of Materials readers as well as descriptions of types of composites, however, the extension of the Introduction is necessary in other parts (see detailed comments). Next, the authors described the material compositions and testing procedures applied during the study. Further, the authors presented results of their studies, which are a subject of numerous doubts and inconsistencies (see detailed comments). After reading the manuscript the goal of preparing and characterization of the obtained materials, at least in the range of the performed testing, remains unclear.

1) The Introduction need to be systematized. In the current form, the authors jump from topic to topic and it is difficult to follow the way of thinking of the authors. It seems that the cited references are not logically connected each other or with the main topics of investigation. The authors discussed manufacturing technologies, characterization, influence of additives, etc., thus, it is better to group these descriptions in particular paragraphs. The style also need to be polished.

2) Since the main topic of the study is copper reinforced thermoplastics for 3D printing, the authors should focus especially on this topic in their literature review, present state-of-the-art, and based on this review, formulate the research problem. Currently, the research problem is not well supported and justified.

3) SI units should be used in the manuscript.

4) It is necessary to provide full information of manufacturers of materials and testing equipment together with specific numbers/models of them to make the study traceable and reproducible.

5) Please comment on which basis the compositions of polymers and copper particles were determined. Does it has a connection with some specific applications?

6) All the variables need to be written with italic font.

7) Line 216: “ν represents the Hook’s Law” – what this statement means? Hooke’s law is represented by the formula, not a single variable.

8) The discussion in section 3.1.1 need to be supported by proofs, i.e. if the authors address to bonding and dimensions of molecules, the microphotographs proving it would be desirable.

9) Due to quite significant differences between the readings of the results presented in tables in section 3, it would be important to determine errors/standard deviations and discuss the reasons of these differences.

10) The discussion on melt flow rate in section 3 need to be extended by analysis of physical and chemical prerequisites that influence of the change of fluidity. Probably, a good parameter to analyze in this study would be the viscosity of the resulting mixtures.

11) Thee specimens for tests seems to be too low value to conclude about properties. According to the standards, usually 5 specimens is the minimum. Looking on the tensile testing results presented in Figures 5-11, it is clear that the differences between specimens are significant, thus, it is recommended to enlarge the number of tested specimens.

12) FTIR results are limited to the description of observations. It is important to discuss the reasons of the observations, for example, appearance of new bonding, etc.

13) According to the obtained results, the general conclusion is that addition of copper particles worsen rheological and mechanical properties of the resulting mixtures compared to a polymer without additives. Since the authors mentioned that the developed materials are dedicated for electronics industry, the electrical and thermal conductivities characterization seem to be important parameters to investigate in this study. Otherwise, the reason of adding copper particles into a polymer is not clear.

14) The language of the manuscript needs substantial revision. Proof-reading is highly recommended.

Reviewer 2 Report

The reviewed manuscript entitled 'Investigations on Rheological and Tensile Behaviour of Single, Double and Triple sized Coppe Reinforced Thermoplastic Composites" concerns the problem of the analysis of the mechanical and physical properties of composite materials reinforced by copper particles of a different size, namely 100 Mesh, 200 Mesh, and 400 Mesh. As the resin, nylon 6, and acrylonitrile butadiene styrene ABS is used. The text of the manuscript consists of four sections. A brief review of the literature and discussion is presented in the introduction. In the next section, the materials and methods are briefly described. In the third section, the results of experiments are presented, namely rheological properties where MFI coefficient for different mixture copper particles and resin are estimated. Next, the mechanical properties are evaluated and, finally, the material characterization (FTIR and SEM) is presented. In the last section, the conclusions are provided.
The manuscript is supplemented by a list of references, which consists of 36 items.

Generally, the manuscript is well written, however, there are some problems that should be corrected.

1. In subsection 2.1 there is clear information according to which standard the MFI studies are performed, namely: 

"ASTM D1238 is just one particular standard that has been used for the Melt Flow Index (MFI) testing [29]."

However, in section 2.2 or 3.2 in the case of estimating the mechanical properties, there is no information about this kind of standard or norms.

2. In sections 2.2 and 3,2 there is no information about the shape of the specimens used for estimating the Peak Load, Peak Elongation, Break Load, Break Elongation, Elongation at Peak, and Elongation
at Break, etc. 

3. From the mechanical point of view the really important quantities are Young modulus and Poisson's ratio of the studied new composite materials. Therefore the Authors should consider adding the estimations of these parameters.

4. Figures 5 - 11 are of rather poor quality. In my opinion, they should be corrected.

Taking under consideration the above flaws I suggest a minor revision of this manuscript.   

Reviewer 3 Report

The paper “Investigations on Rheological and Tensile Behaviour of Single, Double and Triple sized Copper Reinforced Thermoplastic Composites" is focused on a very interesting and useful topic since there are just a few works about these types of composites.  The authors explain fairly well the results and the article is well-written. However, I think that the potential application of these systems is not highlighted. The authors should emphasize this.  

In addition, I think the title is not adequate for this work. Rheology is much more than measuring the Melt Flow Rate. In my opinion, this research is not a rheological investigation and this should be remove from the title because it can lead to confusion.

Round 2

Reviewer 1 Report

The authors of the submitted manuscript investigated the mechanical behavior of thermoplastics reinforced with copper particles for 3D printing applications in electronics. The Introduction can be shortened especially at the beginning, i.e. the description what are the composites seems to be trivial for the community of Materials readers as well as descriptions of types of composites, however, the extension of the Introduction is necessary in other parts (see detailed comments). Next, the authors described the material compositions and testing procedures applied during the study. Further, the authors presented results of their studies, which are a subject of numerous doubts and inconsistencies (see detailed comments). After reading the manuscript the goal of preparing and characterization of the obtained materials, at least in the range of the performed testing, remains unclear.

1) The Introduction still needs adding minor interconnection. The authors did the improvements, however, in every paragraph a kind of introduction to the topic is necessary, e.g. why the authors discuss FDM technique and how it is connected with the composites with metallic particles? Please make appropriate corrections and introduce these interconnections.

2) All the variables need to be written with italic font – please carefully check the manuscript and introduce necessary corrections.

3) Formula (1) was replaced by a set of other formulas, which do not reflect previously discussed phenomena (in version 1 of the submitted manuscript), please comment.

4) It is recommended to replace the SEM images presented in Figure 5 by the images of the resulting composites.

5) If the performing tests for additional specimens is not possible, the authors need to provide a strong argumentation on the obtained results, including the physical prerequisites and literature.

6) The results of mechanical testing clearly show worsening of the mechanical properties when the copper particles are added. It is then essential to focus on other properties discussed in the manuscript, like thermal and electrical conductivity, otherwise, the manuscript does not bring any added value to the problem.
